# The Impact of Internet Use on Land Productivity: Evidence from China Land Economy Survey

**Xiang Deng, Jie Peng and Chunlin Wan \***

School of Economics, Sichuan University, Chengdu 610065, China
* Correspondence: chunlinwan@scu.edu.cn

**Abstract:** Enhancing land productivity is a crucial strategy for addressing key sustainable development issues, such as poverty reduction and ensuring food security. Farmers' Internet use behavior offers the potential to improve land productivity. However, relatively little is known about the association between Internet use and land productivity. To fill this void, this study examines the impact of Internet use on land productivity and its mechanisms. The results indicate that farmers' use of the Internet has a positive impact on improving agricultural land productivity. Internet use increases land productivity by 12.3%, and the conclusion still holds after a series of robustness tests and endogeneity tests. Heterogeneity analysis indicates that Internet use significantly enhances land productivity in the central and northern parts of Jiangsu Province, while it does not have the same effect in the province's southern regions. Without the addition of county fixed effects, the central sample regression results show that the coefficient for Internet use is 0.165 and significant at the 10% confidence level. When county fixed effects are added, the coefficient decreases to 0.117 and is not significant. The coefficient on Internet use for the northern sample is 0.128 and is significant at the 5% confidence level. Mechanistic analyses demonstrate that Internet use also enhances land productivity primarily by expanding the cultivated land area, facilitating mechanized production, and strengthening farmers' social networks. The results of the study indicate that the positive effects of Internet use in improving land productivity should be fully released by strengthening the communication infrastructure, further enhancing farmers' Internet use capacity, improving the land transfer system, upgrading the socialized service level of agricultural machinery, and strengthening agricultural financial support.

**Keywords:** internet use; mechanization; area of land cultivation; social networks; land productivity





## 1. Introduction

The rapid rise in global population has led to a rapid decline in the world's arable land per capita [1], which poses a huge challenge to sustainable development. According to The State of Food Security and Nutrition in the World 2023 released by the Food and Agriculture Organization of the United Nations (FAO), there were 691–783 million people struggling with hunger globally in 2022, which is 122 million more than in 2019 [2]. As one of the most populous countries, China faces an acute demographic-land nexus. China's per capita arable land area in 2021 was 0.08 hectares, which is only 44.4% of the world's [1]. At the same time, the supply of agricultural products in China faces challenges such as high dependence on imports and high production costs, making it unsustainable to rely on traditional methods of expanding inputs and resource consumption to ensure food security. In this context, boosting land productivity and labor productivity is the key measure to address food security and other sustainable development issues.

Internet use provides an opportunity to revolutionize agricultural production methods and increase land productivity. In recent years, the Internet has spread rapidly in many developing countries [3], which greatly accelerates the dissemination of information on agricultural technologies and helps to improve the efficiency of resource allocation in

agricultural production. Theoretically, Internet use can directly and indirectly increase land productivity. On the one hand, through the Internet, farmers can quickly obtain agricultural production information at a lower cost, which helps to improve the efficiency of agricultural resource allocation and management efficiency [4], bringing about an increase in land productivity. For example, if farmers use the Internet, they can efficiently obtain advanced agricultural production technology and guidance at a lower cost to produce with optimal technology and structure. On the other hand, Internet use can indirectly increase land productivity through multiple channels. First, the Internet provides farmers with more information about the job market and helps rural laborers to engage in non-farm employment, which creates conditions for land transfer. At the same time, Internet use can reduce the cost of farmers' land transfer and significantly increase the probability and scale of farmers' land transfer [5]. This will motivate farmers to expand the scale of land cultivation and promote mechanized production. Second, Internet use allows residents of remote areas to break through the physical constraints of the market, which will expand the market for agricultural products and incentivize farmers to update production technology and expand production scale, thereby increasing land productivity [6]. Third, Internet use expands farmers' social networks, which strengthens ties among farmers and facilitates access to financial support from relatives and friends for scaling up production, i.e., Internet use relaxes farmers' credit constraints [7]. Thus, Internet use accelerates farmers' upgrading of production equipment and technology, which will increase land productivity.

Compared with previous studies, the marginal contribution of this paper is reflected in the following three aspects. First, there is little literature discussing the impact of Internet use on land productivity and its mechanism in China, and this study fills the gap. At the same time, this study examines the impact of Internet use on land productivity based on microeconomic survey data, which can, to some extent, avoid the statistical bias and summing errors that may occur in macro data. Second, existing studies disagree on the role of Internet use on land transfers in versus land transfers out. Moreover, it is important whether the positive correlation between the land cultivation scale and land productivity in China remains after land titling, as this relationship determines future agricultural policies. This study enriches the discussion by examining the role of Internet use on land cultivation scale and the relationship between land cultivation scale and land productivity using a mediating variable model. Third, previous studies disagree on the role of Internet use on farm income, which may be due to the fact that they ignore the role of Internet use in land productivity. This study can contribute new insights to related studies from the perspective of land productivity.

The rest of the study is organized as follows: Section 2 is the literature review, Section 3 presents the theoretical analysis and research hypotheses, Section 4 is the research design, Section 5 contains the empirical analysis, Section 6 provides discussion, and Section 7 gives the conclusion and policy recommendations.

## 2. Literature Review

With reference to existing studies [8,9], this paper summarizes the relevant studies from three aspects.

### 2.1. Research on Internet Use

Information Communication Technology (ICT) is one of the fundamental drivers of economic growth and productivity enhancement [10–12]. Internet use has received a great deal of academic attention due to its important role in politics, economy, society, farmers, and agriculture [13]. Most of the literature treats Internet use as a dummy variable and assigns values in terms of whether one has access to broadband or whether one has a cell phone [12,14]. Krueger was the first to estimate the role of Internet use on wages and he found that computer use significantly boosted wages [15]. Subsequent studies have focused on exploring the factors that influence Internet use and the impact of Internet use.

Factors affecting Internet use mainly include personal characteristics and geographical characteristics. In terms of personal characteristics, the more educated an individual is, the more Internet knowledge he possesses and the more likely he is to use the Internet [16]. As individuals become older, the probability of an individual using the Internet decreases. While the elderly have different reasons for not using the Internet, the activities they use the Internet for are similar [17]. Occupation also has an impact on an individual's use of the Internet. For example, salesmen are more likely to use the Internet than individuals working in agriculture because salesmen need to use the Internet to stay in touch with customers [18]. In addition, factors such as household size and labor force share are also positively associated with Internet use [19]. Geographically, the probability of using the Internet is higher for residents of areas that are further away from markets, because using the Internet helps households in remote areas to connect with markets, while reducing communication and transaction costs for households and facilitating the engagement of households in online business activities [16].

Internet use has significant impacts on the economy and society, agriculture and farmers, and poverty reduction. Internet use has growth and distributional effects on the economy and society. Internet use has a positive effect on economic growth in both developed and developing countries. Empirical evidence based on the data from OECD countries from 2002 to 2007 suggests that for every 10% increase in Internet broadband penetration, the annual economic growth rate will increase by 0.25% [20], while a study based on the panel data and instrumental variable method for OECD countries from 1996–2007 finds that for every 10% increase in Internet broadband penetration, the average annual growth rate of GDP per capita would increase by 0.9% to 1.5% [21]. The role of Internet use on economic growth in developing countries is divergent. Estimates based on panel data from 59 countries over the period 1995–2010 suggest that there are no significant differences in the benefits of Internet use in developing, emerging market, and developed countries [22], which is contrary to the conclusion that middle-income and lower-income countries would derive greater benefits from the Internet [23]. The distributional effects of Internet use are manifested in its impact on economic inequality. Individuals with more education can use the Internet to gain more knowledge, which can help to increase the employment opportunities and income of individuals, thus widening the income gap with less educated individuals; in addition, farmers with higher incomes will gain more from Internet use, so Internet use has the potential to exacerbate income inequality among farmers [19]. Internet use has a heterogeneous effect on different age groups, and some studies have focused on the Internet use behavior of the elderly [24,25]. It was found that Internet use helped to increase life satisfaction [26], alleviate loneliness, and reduce depression among the elderly [27].

The role of Internet use on agricultural production is controversial. Empirical evidence based on a survey of farmers in China found that the role of Internet use on agricultural income was not significant and that the Internet was needed to promote the use of new agricultural technologies [28]. In contrast, Internet use has a significant positive effect on Vietnam's total agricultural output, and the effect of Internet use is greater in less developed provinces, as the Internet can provide farmers with more knowledge about agricultural production, such as helping them to improve the efficiency of fertilizers [29]. Although empirical evidence from 81 countries over the period 1995–2000 suggests that emerging communication technologies (ECTs) have a significant positive impact on agricultural productivity, and that differences in the use of ECTs across countries may be one of the important reasons for differences in agricultural productivity [30], the micro-mechanisms that make such differences are still unclear. In terms of production methods, the use of the Internet is conducive to reducing the intensity of the use of chemical fertilizers and promoting the development of green agriculture [28,31]. For farmers, the Internet makes it easier for farmers to access markets and financial services [32,33], promoting the sale of agricultural products and the growth of farmers' income [34]. Estimates based on U.S. panel data from 1999–2007 suggest that broadband availability is strongly correlated

with employment and has a greater impact on rural areas [35]. Overall, there are three mechanisms by which Internet use affects farmers. The first is the information channel, where the Internet gives farmers access to more information and facilitates the learning of new technologies and policies [36]. The second is the human capital channel, where Internet use enhances farmers' human capital [31]. The third is the market channel; Internet use facilitates farmers to connect with the market and provides convenience for the sale of agricultural products [37].

In addition, Internet use plays an important role in poverty reduction. Studies have shown that Internet use has an important dampening effect on multidimensional poverty through credit, social networks, and public services [16]. Developed countries have benefited significantly from Internet use; however, the benefits to developing countries from Internet use are uncertain, particularly in terms of their contribution to poverty reduction [38]. Indeed, Internet use not only increases household incomes and contributes to poverty reduction, but also promotes the development of non-agricultural industries and reduces dependence on natural resources and damage to the environment [19].

### 2.2. Research on Land Productivity

Enhancing land productivity is key to reducing the pressure on ecosystems [39]. When studying agricultural productivity, most of the literature does not strictly distinguish between land productivity and labor productivity [40]. The literature usually defines land productivity as crop revenue or yield per unit of land [41,42]. Actually, land productivity is not the same as labor productivity. Therefore, this study uses crop output per unit of land to measure land productivity. Research on land productivity has focused on discussing the factors that affect land productivity. Earlier literature pointed out that agricultural land productivity has an inverse relationship (hereafter referred to as farm-size IR) with the size of land cultivation when the size of land cultivation exceeds a certain level, i.e., land productivity decreases as farm size increases [43], and this relationship is widely found in many countries [44]. Although some empirical studies have validated this relationship, recent studies have found that this relationship arises from systematic yield misreporting by farmers, i.e., underreporting crop yields on larger plots and overreporting yields on smaller plots [45]. However, data based on maize plots from smallholder farmers in Zambia found the exact opposite phenomenon. The area of smaller plots tends to be overestimated, while the area of larger plots is underestimated, and the inverse relationship between land area and productivity is more pronounced after correcting for errors [46]. In Asian countries, the farm-size IR relationship may be disappearing with economic growth and labor mobility to non-farm sectors [47,48].

China is still at a stage where land productivity rises with the expansion of the scale of land operations, as the degree of land concentration and the scale of land operations by farmers in China are still relatively small [49]. However, the relationship between land productivity and land size in China may have changed after land titling. Most studies have identified labor market failures and monitoring costs as reasons for the formation of farm-size IR [50,51]. Some studies also emphasize the important role of land edge effects or land quality differences, etc., in the formation of farm-size IR [52]. In addition, differences in farmers' occupational choices and skills are equally important factors contributing to differences in land productivity across farmers [53]. Contrary to the scale of land cultivation, mechanization contributes to land productivity. For example, based on data from the 2016 China Labor Force Dynamics Survey, it is shown that semi-mechanized and fully mechanized farming have positive effects on land productivity, with full mechanization having a greater effect. Moreover, mechanization has a greater effect on land productivity of female-headed households [6]. Finally, empirical evidence based on African data suggests that soil quality management, land policy reforms, etc., are also important factors influencing land productivity [54].

*2.3. Research on Internet Use and Land Productivity*

The mismatch of production factors such as land has constrained the upgrading of China's agricultural business practices. Farmers' access to agricultural information through the Internet has greatly facilitated rural land transfer. However, it is controversial to whether Internet use has a greater effect on land transfer in or land transfer out. Some literature argues that Internet use has a greater effect on land transfer in than land transfer out [55]. This is because Internet use enables farmers to efficiently access information related to agricultural production, such as weather, pest control, and so on. It also expands online sales channels and reduces the cost of selling agricultural products [56]. This can stimulate farmers' enthusiasm for production and promote farmers to expand production scale. In contrast, although Internet use improves the market allocation efficiency of agricultural production factors, the effect of farmers' Internet use on the land transfer rate is negative, and the negative effect has a tendency to gradually increase [57,58]. From the perspective of farmland subletting, farmers who use the Internet are more likely to rent out their farmland [13]. This is because the Internet provides farmers with more employment information, which may lead to the flow of farmers into other non-farm industries [40]. According to the farm-size IR relationship, if farmers transfer in land, the increase in farmland area will make land productivity decrease. However, considering that China is still in the stage where land productivity rises with the expansion of land cultivation scale [49], then land productivity will rise. At the same time, there is controversy about the relationship between Internet use and farm income. Empirical evidence based on data from Ghana suggests that Internet use has contributed to the rise in agricultural incomes [59]. However, empirical evidence based on Chinese data found that although Internet use promotes farmers' sustainable production, its effect on farm income is not significant [28]. In fact, when factor inputs are increased, it does not cause an increase in land productivity, although it leads to an increase in farm income.

In summary, previous studies have provided important references for understanding the relationship between Internet use and land productivity. However, little literature has directly examined the relationship between Internet use and land productivity. Meanwhile, the relationship between Internet use and land transfer remains controversial. Whether there is still an inverse relationship between the size of Chinese farmers' land cultivation and land productivity after land titling also remains to be answered. Therefore, there is a huge research space between Internet use and land productivity. Based on this, this study analyzes theoretically and empirically the role of Internet use on land productivity and its influence mechanism based on micro data in China, which helps to fill the gaps of existing studies.

## 3. Theoretical Analysis and Research Hypotheses

The Internet can increase land productivity by providing information such as market information or production techniques [29]. Moreover, farmers with more agricultural production information have higher land returns [60]. The Internet has accelerated the dissemination of information on agricultural science and technology and the promotion of emerging agricultural technologies, which can facilitate the use of the latest agricultural production technologies by farmers and thus improve land productivity. Before agricultural production, the Internet reduces the cost of information collection, and farmers have access to the latest agricultural production technologies through the Internet. Specifically, the Internet provides more up-to-date information on agricultural production tools, which helps to optimize the decision-making and efficiency of farmers' investment in agricultural production. In terms of the choice of types of agricultural products, the Internet can optimize the efficiency of resource allocation by influencing the types of agricultural products that farmers cultivate. For example, by using the Internet, farmers can find agricultural products that are highly profitable, productive, or in short supply. It also helps farmers to answer production questions and influence their production decisions. This improves farmers' land output. In the agricultural production process, Internet use

can promote the digitalization of agricultural production. Specifically, the use of digital technologies, such as sensors and communication technologies, can build a real-time, intelligent management system, which helps to promote the intelligent management of the entire life cycle of agricultural production and reduce the input of production factors like labor and land. At the stage of product sales, the Internet has expanded the sales market for agricultural products, especially shortening the distance between farmers in remote areas and the market, which helps to realize the market value of agricultural products. For example, farmers find suitable buyers through online trading platforms to broaden the marketing of agricultural products, which helps to improve farmers' motivation and farming efficiency [16]. In conclusion, Internet use helps to optimize farmers' production decisions, improve the efficiency of production factors, and stimulate farmers' production motivation, thus increasing the productivity of agricultural land. Based on this, we propose the following hypothesis:

**Hypothesis 1:** *Internet use has a positive effect on land productivity.*

The traditional transfer of agricultural land has problems such as low rent and transactions between acquaintances, which leads to the instability of land transfer and makes it difficult for farmers to efficiently expand the scale of land management. Internet use can facilitate land transfer. As a medium for exchanging information, Internet use can quickly integrate information on land transfers, agricultural activities, etc., which will reduce the cost of information transfer and search for farmers and improve the efficiency of communication among land transfer participants [61]. For example, farmers can obtain more information on land transfer through the Internet platform, which eases the information asymmetry between land transfer participants. At the same time, the record of the Internet platform can effectively control the moral risk after the transaction, which reduces the supervision cost and effectively protects the interests of both parties of land transfer. This will undoubtedly promote the transfer of agricultural land. In addition, the Internet provides a huge amount of information and knowledge. Farmers can learn more about agricultural production and improve their agricultural production skills through the Internet [40], which can improve the efficiency of farmers' large-scale production. The increase in land size has an economy of scale effect because it promotes the mechanization of agricultural production, which reduces the inputs of factors of production, thereby reducing the unit cost of production and contributing to higher land productivity. This leads to the following hypothesis:

**Hypothesis 2:** *The scale of land cultivation plays a mediating role. Internet use can increase land productivity by expanding the area of land cultivated by farmers.*

Internet use has facilitated the transfer of surplus rural labor and led to non-farm employment, which has contributed to the mechanization of agricultural production. The Internet platform provides farmers with abundant information on non-farm employment, which not only effectively reduces individual information search costs but also alleviates the information asymmetry problem in the traditional labor market. Internet use increases the probability of non-farm employment for farmers. The significant seasonal nature of agricultural labor and the fact that non-farm employment raises the opportunity cost of agricultural labor make the choice of mechanized production a natural and optimal alternative to agricultural labor [62]. Farmers can mechanize their production by purchasing farm machinery or by purchasing farm machinery services; however, the cost of purchasing farm machinery is high because of the high threshold for mechanization and the small size of farmers' land. Internet use has increased the availability of agricultural machinery services. The Internet enhances communication between farmers and helps them to choose appropriate farm machinery services, which increases the willingness to purchase farm machinery services. Therefore, Internet use helps to promote farm mechanization and thus increase land productivity. Based on this, this study proposes the following hypothesis:

**Hypothesis 3:** *Mechanized production plays a mediating role. Internet use can increase land productivity by facilitating mechanized production.*

Internet use can enhance interaction among farmers and accelerate the dissemination of knowledge related to agricultural production [63]. The Internet has broken the distance limitation of people's interaction and reshaped the social networks structure of farmers. By using the Internet, farmers not only strengthen their ties with friends and relatives, but also expand their existing social networks. The new social networks will undoubtedly contribute to increased land productivity. First, the Internet can accelerate the flow of agricultural production technology among farmers, reduce the cost of learning new knowledge and technology for farmers, and promote the spillover effect of agricultural technology. Second, Internet use can ease the credit constraints of farmers. The expansion of social networks has made it less difficult for farmers to access finance, which can help them to upgrade production equipment and make long-term investments. Third, the expansion of social networks helps farmers to access production services at lower costs and lowers the threshold for farmers to use advanced production technologies. At the same time, the Internet can broaden the market for agricultural products. This helps to increase the land productivity of farmers. Based on this, this study proposes the following hypothesis:

**Hypothesis 4:** *Social network relationships play a mediating role. The use of the Internet helps to strengthen the social network relationships of farmers, thereby increasing land productivity.*

## 4. Research Design

### 4.1. Study Aera

The data for this study come from China's Jiangsu Province, which is in the Yangtze River Delta region, on the eastern coast of mainland China, with an area of 107,200 square kilometers. There are four reasons for choosing Jiangsu Province as the study area in this paper. First, Jiangsu Province is dominated by plains, which account for 86.89% of the land area, ranking first among all provinces in China. Jiangsu Province is also an important grain production base in China, with a grain cultivation area of 81.67 million mu and an output of 37.69 million tons in 2022. Therefore, Jiangsu Province is an excellent choice for studying land productivity. Second, Jiangsu Province has a well-developed economy, and all its prefecture-level cities are among the top 100 prefecture-level cities in China. Jiangsu Province is leading in the construction of Internet infrastructure. By the end of 2021, the total length of fiber optic cables in the province was 4.16 million kilometers, ranking the first in China, and the number of Internet users reached 65.66 million, of which cell phone Internet users accounted for 99.6% and rural Internet users accounted for 24.4%. Considering that the core explanatory variable in this paper is Internet use, this helps to minimize the error associated with poor infrastructure development. Third, Jiangsu Province can usually be divided into three regions, which helps to test for possible heterogeneity. Fourth, the Social Sciences Division of Nanjing Agricultural University created the China Land Economy Survey in 2020, which covers all of Jiangsu Province, and this database provides a suitable data source for studying land productivity.

As shown in Figure 1, the figure shows the relative geographic location of Jiangsu Province and the proportion of farmers using the Internet in each city in the survey data. The proportion of farmers using the Internet in the survey data is 0.454, indicating that the overall proportion of farmers using the Internet in Jiangsu Province is relatively low, and there is still much room for improvement. In terms of cities, Suzhou, Yancheng, and Huai'an have the highest proportion of farmers using the Internet. Specifically, the proportion of farmers using the Internet in Suzhou is 60.5%, ranking first among all cities. This is because Suzhou has a well-developed economy and farmers receive more education. According to the survey data used in this study, the average year of education of surveyed farmers in Suzhou is 7.765 years, which is 9.044% higher than the average year of education of all surveyed farmers in Jiangsu Province (7.121 years). Lianyungang, Yangzhou, and

Taizhou are at the bottom of the list in terms of the proportion of farm households using the Internet. The percentage of farm households using the Internet in Lianyungang is only 0.266, ranking last among all cities, which may be due to the city's relatively poor economic development and farmers' relatively short years of education. According to the survey data of this study, the average length of education of Lianyungang farmers is 6.575 years, which is only 92.333% of all surveyed farmers in Jiangsu Province. In addition, villages in Lianyungang had the lowest average income in 2019, which may have affected the Internet use behavior of farmers.

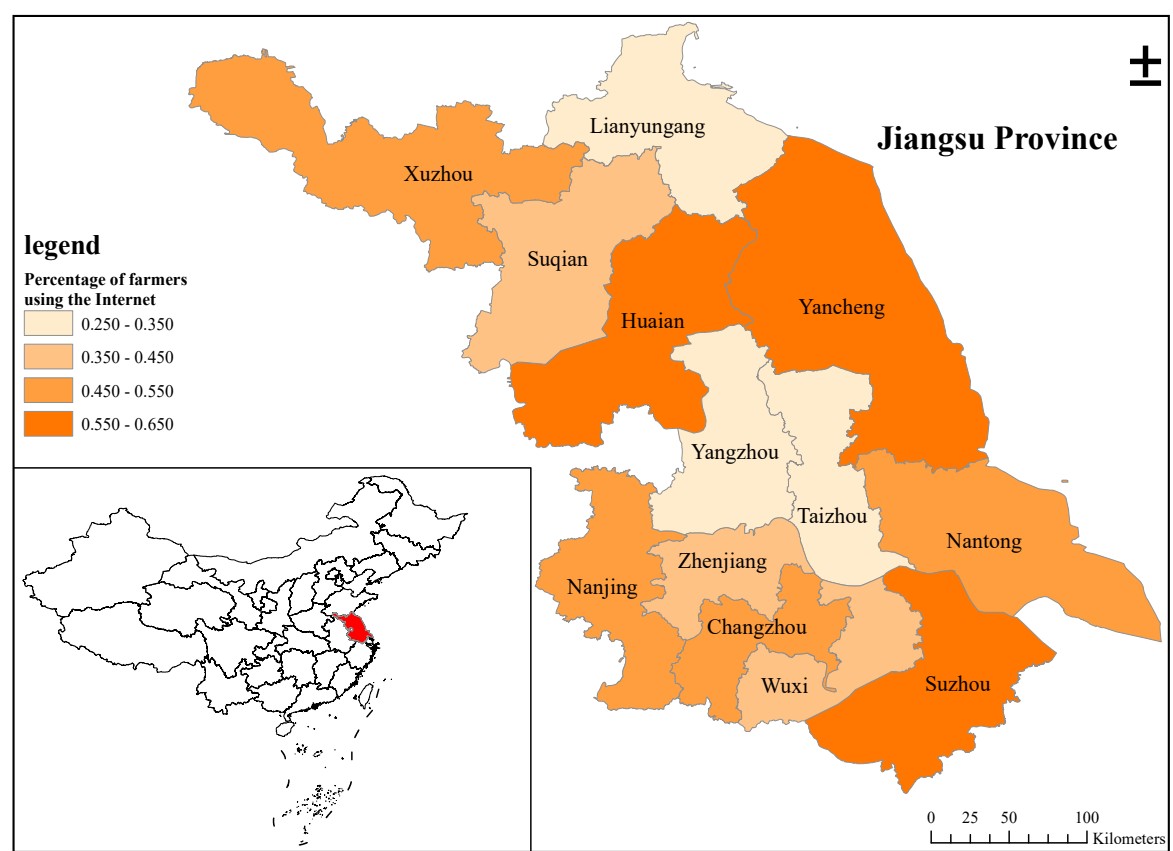

**Figure 1.** Relative geographic location of Jiangsu Province, China.

According to economic development, geographic location, and other conditions, Jiangsu Province can be divided into three regions: North Jiangsu Province, Central Jiangsu Province, and South Jiangsu Province. The prefecture-level cities included in each region are shown in Table 1. According to the survey data, there is an obvious gap between the Internet usage rates of farmers in the three regions. As shown in Figure 1, the mean value of Internet usage rate is the highest in Southern Jiangsu Province, followed by Northern Jiangsu Province, and the mean value of Internet usage rate of surveyed farmers in Central Jiangsu Province is the lowest.

**Table 1.** Cities included in the three regions of Jiangsu Province.

| Region of Jiangsu Province | City |
| --- | --- |
| South Jiangsu Province | Suzhou, Wuxi, Changzhou, Zhenjiang, Nanjing |
| Central Jiangsu Province | Yangzhou, Taizhou, Nantong |
| North Jiangsu Province | Xuzhou, Lianyungang, Huai'an, Yancheng, Suqian |

### 4.2. Data Sources

The data used in this study came from the China Land Economy Survey (CLES), Nanjing Agricultural University. The database provides a data cleaning report, a data review report, a village questionnaire, and a farm household questionnaire. The farm household questionnaire and the village questionnaire are the main sources of data, where the farm household questionnaire includes questions on 11 aspects such as basic information, income, land use, and parcel information. The village questionnaire includes questions on 10 aspects such as basic information, demographic characteristics, and land use and transfer.

In 2020, CLES completed a baseline study that covered 13 prefecture-level cities throughout Jiangsu Province. First, two counties were sampled from each prefecture-level city. Second, two villages from each county were selected for the survey, yielding a total of 52 administrative villages. Finally, using random sampling, 50 farm households were selected from the sample villages, resulting in 2600 farm households. In 2021, the CLES project conducted a follow-up survey. Due to the impact of COVID-19, tracking surveys were completed in only 12 municipalities. The average tracking rate of the survey was 63.8%, and those farmers who were not tracked were supplemented using other farmers in the same village. In 2022, CLES completed a follow-up survey in six municipalities. To include as many samples as possible and improve the accuracy of the findings, this study chose the baseline data of 2020 for empirical research, and 1581 valid samples were obtained after data processing.

### 4.3. Variable Selection

The core dependent variable in this study is land productivity. Referring to the practice in the literature [6,64], we use crop yield per acre to represent land production efficiency. It is obtained from the ratio of crop yield to planted area. To reduce the effect of heteroskedasticity, we use the natural logarithm of this ratio to characterize agricultural production efficiency. In the robustness test, considering that rice is the most important agricultural product in Jiangsu, we use the logarithm of rice yield per acre as a proxy variable for the explained variable. At the same time, referring to existing studies [49], we also use the logarithm of agricultural yield per capita as a proxy variable for land productivity to enhance the robustness of the findings.

The core explanatory variable is Internet use. Referring to existing studies [12,14], this variable is a dummy variable that is assigned a value based on whether the question "If you go online, what is your primary method of accessing the Internet (multiple choice)" is answered. If the question was answered, it is 1, if not, it is 0.

There are three mediating variables. They are mechanized production, area of land cultivation, and social networks. Mechanized production is measured by the cost of machinery operations per capita [65]. First, we multiply the area under cultivation by the cost of machinery per acre to obtain the total cost of machinery. Then, we can obtain the ratio of the total cost of machinery operations to the resident population. Finally, we take the natural logarithm of this ratio to obtain a proxy variable for mechanized production. We measure the size of land cultivation by the natural logarithm of the total area cultivated by the farm household. Social networks can be represented by the natural logarithm of interpersonal expenditures in household expenditures.

To minimize the effect of omitted variables on the regression results, we introduced three types of control variables [5,42,49]. The first category is the household head characteristics variables, including age, gender, education, health status, and marital status of the head of the household. The second category is household characteristics variables, including household size, whether there is a cadre in the household, whether there is a party member in the household, the average age of the household, the Proportion with agricultural education or training, the time invested in agricultural labor, the health status of the members, proportion with non-agricultural education or training, and whether the household has access to agricultural credit services. The third category is village characteristics variables, including economic development of villages and distance from village

council to township government. Specific definitions and descriptive statistics for each variable are shown in Table 2.

**Table 2.** Variable definition and descriptive statistics.

| Variable | Variable Name | Variable Definition | Mean | S.D. |
|---|---|---|---|---|
| Dependent Variable | Agricultural land productivity (eff) | Natural logarithm of crop yield per acre | 7.262 | 0.930 |
| Independent Variable | Internet Use (Int) | If you use the Internet, the main way you access the Internet is (multiple choice)? 1 = Answer to the question, 0 = No answer to the question | 0.446 | 0.497 |
| Mediating variables | Mechanized production (pmcst) | Natural logarithm of per capita cost of machinery production for the resident population | 5.961 | 1.719 |
| | Area of land cultivation (lands) | Natural logarithm of total area planted by farmers | 1.986 | 1.681 |
| | Social networks (rltion) | Natural logarithm of interpersonal expenditures in household expenditures | 8.366 | 0.930 |
| Control variables | Age | Age of the head of household in 2020 | 61.745 | 10.178 |
| | Gender | Gender of the head of household: 1 = male, 0 = female | 0.924 | 0.264 |
| | Education (edu) | Education years of the head of household | 7.172 | 3.555 |
| | Health (heal) | Health status of the head of household: 1 = incapacitated; 2 = very unhealthy; 3 = relatively unhealthy; 4 = relatively healthy; 5 = very healthy | 3.904 | 1.071 |
| | Marital status (marr) | Whether the head of household is married: 1 = married; 0 = unmarried | 0.909 | 0.288 |
| | Household size (popu) | Number of persons in the household | 4.385 | 1.897 |
| | Cadre (cad) | Is there a cadre in your family? 1 = Yes; 0 = No | 0.136 | 0.343 |
| | Party member (part) | Is there a party member in your family? 1 = Yes; 0 = No | 0.276 | 0.447 |
| | Average age (aage) | Average age of household members in 2020 | 47.794 | 12.721 |
| | Proportion with agricultural education or training (aratio) | Population with agricultural education or training divided by household size | 0.166 | 0.272 |
| | Agricultural labor inputs (agd) | Agricultural Labor Days for the Whole Family in 2019 | 4.573 | 1.080 |
| | Health status of family members (healrt) | Percentage of healthy family members | 0.784 | 0.292 |
| | Proportion with non-agricultural education or training (naratio) | Population with non-agricultural education or training divided by household size | 0.126 | 0.211 |
| | agricultural credit services (agcred) | Have you accessed agricultural credit services? 1 = Yes; 0 = No | 0.007 | 0.083 |
| | Economic development of villages (vainc) | Per capita income of the village | 9.783 | 0.559 |
| | Distance from village council to township government (dist) | Distance from village council to township government (kilometers) | 6.656 | 6.962 |

### 4.4. Empirical Methods

4.4.1. Benchmark Regression

To analyze the impact of Internet use on land productivity, we refer to the literature studying land productivity [64,66] and construct the following OLS benchmark regression model:

$$\text{eff}_i = \alpha_0 + \beta_0 \text{net}_i + \sum_j \beta_j \text{Controls}_{ij} + \mu_i \tag{1}$$

where $\text{eff}_i$ denotes the land productivity of the ith farmer, $\text{net}_i$ denotes the dummy variable indicating whether the ith farmer uses the Internet, $\text{Controls}_{ij}$ denotes the jth control variable of the ith farmer, $\alpha_0$ is the constant term, $\beta_0$ is the regression coefficient of the core explanatory variable $\text{net}_i$, $\beta_j$ is the regression coefficient of the control variables, and $\mu_i$ is a random disturbance term that follows normal distribution.

4.4.2. Stochastic Frontier Production Function

To test the robustness of the benchmark regression results, this study replaced the land productivity measure and regressed the results as explained variable using the benchmark regression model. Compared with the nonparametric data envelopment analysis, the stochastic frontier function can fully consider the impact of stochastic factors on output. Therefore, this study uses the stochastic frontier production function to estimate land productivity and uses it for robustness testing. Given the more flexible form of the beyond logarithmic production function and allowing for a nonlinear relationship between input factors and outputs, this paper sets a stochastic frontier production function as shown in Equation (2) to estimate land productivity.

$$
\begin{aligned}
\ln Y_i = {}& \alpha_1 + \beta_L \ln L_i + \beta_K \ln K_i + \beta_D \ln D_i + \beta_{LK} \ln L_i \ln K_i + \\
& \beta_{LD} \ln L_i \ln D_i + \beta_{KD} \ln K_i \ln D_i + 1/2\beta_{LL}(\ln L_i)^2 + \\
& 1/2\beta_{KK}(\ln K_i)^2 + 1/2\beta_{DD}(\ln D_i)^2 + v_i + \theta_i
\end{aligned}
\tag{2}
$$

where $Y_i$ is the total crop yield of household i, $\alpha_1$ is a constant term, L, K, and D are the labor, capital, and land input, respectively, $\beta$ is the coefficient of the primary, interaction, and quadratic terms of each factor, $v_i$ is a random error term following normal distribution, and $\theta_i$ is the loss of efficiency, which represents the degree of deviation of farmer i's land productivity relative to the production frontier.

4.4.3. Mediation Effect Model

This study uses a mediation effect model to test the mechanism of the impact of Internet use on land productivity. First, we construct the regression model as shown in Equations (3) and (4). Second, based on Equation (3), we test the effect of Internet use on the three mechanism variables separately. Finally, we use Equation (4) to test the effect of Internet use on land productivity through the three mechanism variables.

$$
M_i = \alpha_2 + \gamma_0 \text{net}_i + \sum_j \gamma_j \text{Controls}_{ij} + \delta_i
\tag{3}
$$

$$
\text{eff}_i = \alpha_3 + {} + \alpha_4 M_i + \eta_0 \text{net}_i + \sum_j \eta_j \text{Controls}_{ij} + \varepsilon_i
\tag{4}
$$

where $M_i$ is the mechanism variable, $\alpha_2$, $\alpha_3$ are constant terms, $\alpha_4$ is the regression coefficient of the mechanism variable, $\gamma_0$, $\eta_0$ are the key parameters to be estimated. $\delta_i$ and $\varepsilon_i$ are random error terms following normal distribution.

**5. Empirical Analysis**

*5.1. Benchmark Model Regression Results*

Table 3 presents the results of the baseline regression of Internet use affecting land productivity. One can obtain the following information from the table. First, as shown in column (1), the regression coefficient for Internet use is 0.091 when fixed county effects and control variables are not included and it is significant at the 10% confidence level, which suggests that Internet use contributes to land productivity. Second, as shown in columns (2) and (3), the regression coefficients of Internet use on land productivity become larger with the inclusion of fixed county effects or control variables, at 0.114 and 0.123, respectively. They are both significant at the 5% confidence level. Third, as shown in column (4), the regression coefficient for Internet use is 0.123 and passes the significance test at the 1% confidence level after including both fixed county effects and control variables. In

summary, Internet use does contribute to land productivity, which is similar to the findings of Twumasi et al. [67]. Farmers who use the Internet have higher land productivity and Hypothesis 1 is tested.

**Table 3.** Estimated results of the impact of Internet use on land productivity.

| Variable | (1) | (2) | (3) | (4) |
|---|---|---|---|---|
| | eff | eff | eff | eff |
| net (computing) | 0.091 * | 0.114 ** | 0.123 ** | 0.123 *** |
| | (0.047) | (0.045) | (0.052) | (0.045) |
| Constant | 7.221 *** | 6.159 *** | 9.703 *** | 7.663 *** |
| | (0.031) | (0.128) | (0.467) | (0.636) |
| Fixed county | No | Yes | No | Yes |
| Control variables | No | No | Yes | Yes |
| Observations | 1581 | 1581 | 1529 | 1516 |
| R-squared | 0.002 | 0.168 | 0.054 | 0.188 |

Note: *** $p < 0.01$, ** $p < 0.05$, * $p < 0.1$, standard deviation in parentheses. Source: Authors' calculations, the same below.

### 5.2. Robustness Test

To improve the robustness of our findings, we conducted a series of robustness tests, including replacing the explanatory variables, replacing the core explanatory variables, and replacing the land productivity measure.

### 5.2.1. Replacement of Dependent Variable

The regression results after replacing the dependent variable are shown in Table 4. The dependent variables in columns (1)–(4) are the natural logarithm of rice yield per acre. Where regression (1) does not include any control variables, the regression coefficient for Internet use is 0.114 and it is significant at the 5% confidence level. This is consistent with the conclusion of the benchmark regression that Internet use contributes to land productivity. Columns (2) and (3) include fixed county effects and control variables, respectively. The results of both regressions show that the coefficient of Internet use is significantly positive. Regression (4) adds both fixed county effects and control variables, and the regression coefficient for Internet use becomes smaller, but it is still significantly positive at the 10% confidence level. The dependent variables in columns (5)–(8) are the natural logarithm of agricultural yield per capita, where population refers to the permanent population. Regression (5) does not introduce any control variables, the coefficient for Internet use is 0.581, and it is significant at least at the 1% confidence level. Regressions (6) and (7) introduce fixed county effects or control variables, respectively, and the regression coefficient for Internet use in both regressions is significant at the 1% confidence level. Column (8) introduces both fixed county effects and other control variables, and the regression coefficient for Internet use becomes smaller, but it is still significant at the 1% confidence level. The above results suggest that Internet use contributes to land productivity. The conclusions of the benchmark regression are robust and Hypothesis 1 is confirmed again.

**Table 4.** Robustness test results of replacing dependent variable.

| Variable | (1) | (2) | (3) | (4) | (5) | (6) | (7) | (8) |
|---|---|---|---|---|---|---|---|---|
| | eff1 | eff1 | eff1 | eff1 | eff2 | eff2 | eff2 | eff2 |
| net (computing) | 0.114 ** | 0.116 *** | 0.104 ** | 0.086 * | 0.581 *** | 0.468 *** | 0.327 *** | 0.235 *** |
| | (0.045) | (0.043) | (0.049) | (0.047) | (0.094) | (0.084) | (0.093) | (0.083) |
| Constant | 6.693 *** | 5.829 *** | 8.478 *** | 7.028 *** | 7.462 *** | 6.714 *** | 11.155 *** | 4.992 *** |
| | (0.030) | (0.135) | (0.429) | (0.656) | (0.063) | (0.240) | (0.837) | (1.191) |
| Fixed county | No | Yes | No | Yes | No | Yes | No | Yes |

**Table 4.** *Cont.*

| Variable | (1) | (2) | (3) | (4) | (5) | (6) | (7) | (8) |
|---|---|---|---|---|---|---|---|---|
| | eff1 | eff1 | eff1 | eff1 | eff2 | eff2 | eff2 | eff2 |
| Control variables | No | No | Yes | Yes | No | No | Yes | Yes |
| Observations | 1293 | 1293 | 1251 | 1251 | 1642 | 1642 | 1579 | 1579 |
| R-squared | 0.005 | 0.165 | 0.044 | 0.177 | 0.023 | 0.282 | 0.256 | 0.437 |

Note: *** $p < 0.01$, ** $p < 0.05$, * $p < 0.1$.

### 5.2.2. Replacement of Core Independent Variable

The regression results of replacing the core independent variable are shown in Table 5. The core independent variable in columns (1)–(2) is the number of smartphones in the household (nubph). Column (1) introduces only the control variables, the regression coefficient for nubph is 0.042 and it is significant at the 5% confidence level. Column (2) introduces both fixed county effects and all control variables, and one can see that the regression coefficient for nubph becomes smaller, but it is still significantly positive at the 5% confidence level. The more smartphones the household has, the more the Internet affects the farmer and thus the more it helps to increase land productivity, as the number of smartphones reflects the Internet use of the farmer. Therefore, the regression coefficient of nubph is positive, indicating that Internet use helps to enhance land productivity. The core independent variable in columns (3)–(4) is the dummy variable nubcp, which is assigned a value based on the answer to the question "Do you have a computer with Internet access?". If the farmer answers "yes", nubcp takes the value of 1. If the answer is "no", nubcp takes the value of 0. The regression results show that the coefficient on nubcp is positive and significant at least at the 10% confidence level, regardless of whether fixed county effects are included. This indicates that having computer with Internet access at home has a significant positive effect on land productivity, i.e., Internet use contributes to land productivity. The core independent variable in columns (5)–(6) is the number of smartphones per capita (avnph). Column (5) does not introduce fixed county effects. The coefficient on avnph is 0.083 and is significant at the 5% confidence level. Column (6) incorporates both fixed county effects and control variables. avnph's regression coefficient is reduced, but it is still significant at the 10% confidence level. This suggests that the higher the number of smartphones per farmer, the higher their land productivity, i.e., Internet use contributes to land productivity. In summary, the results of the benchmark regression are highly robust and Hypothesis 1 is confirmed again.

**Table 5.** Robustness test results of replacing core independent variable.

| Variable | (1) | (2) | (3) | (4) | (5) | (6) |
|---|---|---|---|---|---|---|
| | eff | eff | eff | eff | eff | eff |
| nubph | 0.042 ** | 0.037 ** | | | | |
| | (0.019) | (0.018) | | | | |
| nubcp | | | 0.091 * | 0.139 *** | | |
| | | | (0.053) | (0.053) | | |
| avnph | | | | | 0.083 ** | 0.056 * |
| | | | | | (0.034) | (0.033) |
| Constant | 9.894 *** | 7.590 *** | 9.816 *** | 7.791 *** | 9.832 *** | 7.584 *** |
| | (0.477) | (0.715) | (0.479) | (0.731) | (0.477) | (0.716) |
| Control variables | Yes | Yes | Yes | Yes | Yes | Yes |
| Fixed county | No | Yes | No | Yes | No | Yes |
| Observations | 1503 | 1503 | 1297 | 1297 | 1503 | 1503 |
| R-squared | 0.056 | 0.199 | 0.057 | 0.187 | 0.056 | 0.198 |

Note: *** $p < 0.01$, ** $p < 0.05$, * $p < 0.1$.

### 5.2.3. Replacement of Land Productivity Measures

First, based on Equation (2), we use the stochastic frontier production function to measure land productivity (effsf). Second, we regress effsf on each of the four proxies for the core independent variables. Finally, we obtain the regression results as shown in Table 6. All regressions in the table incorporate control variables. The core independent variable in columns (1)–(2) is Internet use. Regression (1) without fixed county effects has a coefficient of 0.008 for Internet use and the regression coefficient is significant at the 5% confidence level. After adding fixed county effects, the coefficient on Internet use in regression (2) decreases to 0.007, which is significant at the 10% confidence level. The core independent variable in columns (3)–(4) is nubph. Before controlling for fixed county effects, the regression coefficient for nubph is 0.002 and is significant at the 5% confidence level. After controlling for fixed county effects, the regression coefficient for nubph decreases to 0.001, which is significant at the 10 percent confidence level. The core independent variable in columns (5)–(6) is nubcp. The regression coefficients for nubcp are all positive and significant at least at the 10% confidence level whether the fixed county effects are introduced. The core independent variable in columns (7)–(8) is avnph. We can see that the regression coefficients for avnph are all positive and significant at the 5% and 10% confidence levels, respectively. In summary, we use land productivity obtained from the stochastic frontier production function as dependent variable and finally find that the regression results also support Hypothesis 1.

**Table 6.** Robustness test results of replacing land productivity measures.

| Variable | (1) | (2) | (3) | (4) | (5) | (6) | (7) | (8) |
|---|---|---|---|---|---|---|---|---|
| | effsf | effsf | effsf | effsf | effsf | effsf | effsf | effsf |
| net (computing) | 0.008 ** (0.004) | 0.007 * (0.004) | | | | | | |
| nubph | | | 0.002 ** (0.001) | 0.001 * (0.001) | | | | |
| nubcp | | | | | 0.007 * (0.004) | 0.008 * (0.004) | | |
| netrt | | | | | | | 0.005 ** (0.002) | 0.003 * (0.002) |
| Constant | 1.056 *** (0.034) | 0.994 *** (0.054) | 1.000 *** (0.020) | 1.078 *** (0.031) | 1.075 *** (0.036) | 1.001 *** (0.057) | 1.038 *** (0.025) | 1.008 *** (0.039) |
| Control variables | Yes | Yes | Yes | Yes | Yes | Yes | Yes | Yes |
| Fixed county | No | Yes | No | Yes | No | Yes | No | Yes |
| Observations | 1438 | 1438 | 1198 | 1198 | 1216 | 1216 | 1352 | 1352 |
| R-squared | 0.035 | 0.110 | 0.046 | 0.155 | 0.032 | 0.124 | 0.046 | 0.117 |

Note: *** $p < 0.01$, ** $p < 0.05$, * $p < 0.1$.

### 5.3. Heterogeneity Analysis

In this study, we further analyze the heterogeneous effects of Internet use on land productivity from the perspective of the three major regions of Jiangsu Province, and the regression results are shown in Table 7. Columns (1)–(2) of the table present the regression results for the sample from southern Jiangsu Province. Column (1) does not include fixed county effects, and we find that the coefficient for Internet use is -0.144 and insignificant. Column (2) includes both control variables and fixed county effects, and the absolute value of the regression coefficient is reduced, but still insignificant. The regression results using the sample from central Jiangsu Province are shown in columns (3)–(4). The regression coefficient for Internet use in column (3) is 0.165 when fixed county effects are not included and is significant at the 10 percent confidence level. The regression coefficient of net in column (4) decreases after adding fixed county effects, but it is not significant. The regression results for northern Jiangsu Province show that the regression coefficient of net in column (5) is 0.135 and significant at the 5% confidence level when fixed county effects

are not controlled. After controlling for fixed county effects, the regression coefficient of net in column (6) decreases to 0.128 and remains significant at the 5% confidence level. It suggests that Internet use helps to improve land productivity in the northern part of Jiangsu Province. In summary, from the perspective of the three major regions in Jiangsu Province, there is obvious regional heterogeneity in the impact of Internet use on land production efficiency. It is mainly manifested in the enhancing effect of Internet use on land productivity of farmers in the central and northern parts of Jiangsu Province.

**Table 7.** Results of the heterogeneity test.

| Variant | Southern Part | | Central Section | | Northern Part | |
|---|---|---|---|---|---|---|
| | **(1)** | **(2)** | **(3)** | **(4)** | **(5)** | **(6)** |
| | eff | eff | eff | eff | eff | eff |
| net (computing) | −0.144 | −0.086 | 0.165 * | 0.117 | 0.135 ** | 0.128 ** |
| | (0.154) | (0.145) | (0.099) | (0.092) | (0.059) | (0.059) |
| Constant | 13.084 *** | 13.173 *** | 9.587 *** | 8.050 *** | 8.403 *** | 8.333 *** |
| | (2.927) | (4.448) | (1.191) | (1.359) | (0.489) | (0.675) |
| Control variables | Yes | Yes | Yes | Yes | Yes | Yes |
| Fixed county | No | Yes | No | Yes | No | Yes |
| Observations | 331 | 331 | 410 | 410 | 788 | 788 |
| R-squared | 0.110 | 0.268 | 0.105 | 0.247 | 0.047 | 0.096 |

Note: *** $p < 0.01$, ** $p < 0.05$, * $p < 0.1$.

The regional heterogeneity as shown above may be due to the high rate of urbanization in the southern part of Jiangsu Province, whose economic development is dominated by secondary and tertiary industries, while agriculture is not the preferred industry in the region. For example, in 2019, the urbanization rates of the southern, central, and northern regions of Jiangsu Province were 77.6%, 67.8%, and 64.4%, respectively, and the urbanization rate of the southern region was substantially higher than that of the other two regions. Although the southern region of Jiangsu Province contains five cities, there are only 331 valid regression samples. Meanwhile, as shown in Table 8, the grain production share of the three major regions of Jiangsu Province in 2019–2022 shows that the southern region of Jiangsu Province does not account for more than 11% of grain in any one year, and the average value is only 10.889%. This is substantially lower than the average value of the central and northern regions of Jiangsu Province, which are 24.284% and 64.827%, respectively. As a result, the impact of Internet use on farm households is difficult to be reflected in agricultural production, leading to a non-significant effect of Internet use on land productivity.

**Table 8.** Grain Yield Share of Three Major Regions in Jiangsu Province, 2019–2022.

| Area | 2019 | 2020 | 2021 | 2022 | Mean |
|---|---|---|---|---|---|
| South Jiangsu Province | 10.887% | 10.783% | 10.929% | 10.958% | 10.889% |
| Central Jiangsu Province | 24.416% | 24.337% | 24.240% | 24.141% | 24.284% |
| North Jiangsu Province | 64.697% | 64.880% | 64.831% | 64.901% | 64.827% |

Source: Calculations based on data from the Department of Agriculture and Rural Development of Jiangsu Province.

### 5.4. Endogenous Test

The use of OLS estimation may have endogeneity problems. The first problem is omitted variables. Due to the large number of factors affecting land productivity, even though we have controlled for farmers' individual characteristics, household characteristics, village characteristics, and county differences, the regression may still omit explanatory variables, which will lead to biased estimates. The second issue is inverse causality. Farmers with better agricultural production techniques tend to be more productive on their land,

which increases their income and thus promotes the use of the Internet. The third problem is sample selection bias. Farmers' Internet use behavior and agricultural land productivity may be simultaneously affected by unobservable factors. Referring to an existing study [68], we use the instrumental variable method to estimate the impact of Internet use on land productivity again. The instrumental variable we choose is the Internet usage rate of the farmer's village, which excludes this farmer. On the one hand, the Internet usage rate of the farmer's village reflects the villagers' motivation to use the Internet, which will affect that farmer's propensity to use the Internet. On the other hand, since this farmer is excluded, the village Internet usage rate does not have a direct impact on the land productivity of this individual.

Table 9 presents the estimation results using instrumental variables. The dependent variables in columns (1)–(3) are the natural logarithm of crop production per acre, the natural logarithm of rice production per acre, and the logarithm of agricultural production per capita, respectively. The regression coefficients for Internet use are significantly positive at the 1%, 10%, and 1% confidence levels, respectively. Meanwhile, the Kleibergen–Paap rk LM for all three regressions passes the non-identifiability test at least at the 1% confidence level. Apart from regression (2), the Kleibergen–Paap rk Wald F is greater than the 10% critical value. In short, it can be assumed that there is no weak instrumental variable problem. In summary, after accounting for endogeneity, the effect of Internet use on land productivity remains significantly positive, and Hypothesis 1 is confirmed again.

**Table 9.** Endogeneity test results.

| Variable | (1) | (2) | (3) |
|---|---|---|---|
| | eff | eff1 | eff2 |
| net (computing) | 1.964 *** | 1.107 * | 4.747 *** |
| | (0.686) | (0.646) | (1.277) |
| Constant | 4.228 *** | 5.007 *** | −2.408 |
| | (1.071) | (0.953) | (2.104) |
| Fixed county | Yes | Yes | Yes |
| Control variables | Yes | Yes | Yes |
| Kleibergen–Paap rk LM | 20.379 {0.000} | 8.637 {0.003} | 20.568 {0.000} |
| Kleibergen–Paap rk Wald F | 20.831 [16.38] | 8.616 [16.38] | 20.991 [16.38] |
| Observations | 2743 | 2269 | 3028 |

Note: *** $p < 0.01$, * $p < 0.1$. {} is the *p*-value and [] is the 10% threshold for the instrumental variable weak identification test.

### 5.5. Mechanism Test

The results above indicate that farmers' Internet use behavior has a significant positive effect on land productivity. In order to further test the mechanism of the role of Internet use in enhancing land productivity and to test hypotheses 2, 3, and 4, we conducted the following tests. We tested for mediating effects based on Equations (3) and (4) in terms of mechanized production, land operation scale, and social networks, respectively. The results of the mechanistic tests are shown in Table 10. Columns (1), (3), and (5) report the results of the regressions of Internet use on per capita mechanical operation costs, land cultivation area, and interpersonal expenditures, respectively. The coefficients of Internet use in all three regressions are significantly positive. This indicates that Internet use promotes mechanized production, expands the area of land management, and strengthens the social networks of farmers. The regression results after adding the mediating variables are shown in columns (2), (4), and (6). The coefficients of the mechanism variables in all three regressions are significantly positive. The regression coefficients for Internet use are 0.089, 0.088, and 0.116, and they are significant at the 5%, 10%, and 1% confidence levels, respectively. Meanwhile, in column (4) of Table 3, the coefficient for Internet use is 0.123 and this coefficient is significant at the 1% confidence level. It is larger than the regression coefficient in the mechanism test and it is more significant. Therefore, Internet use can

enhance land productivity by promoting mechanized production, facilitating farmers to expand land cultivation area, and enhancing farmers' social networks. Hypotheses 2, 3, and 4 are verified.

**Table 10.** Mechanism test results.

| Variable | (1) | (2) | (3) | (4) | (5) | (6) |
|---|---|---|---|---|---|---|
| | pmcst | eff | lands | eff | rltion | eff |
| net (computing) | 0.162 * | 0.089 ** | 0.187 *** | 0.088 * | 0.217 *** | 0.116 *** |
| | (0.092) | (0.043) | (0.069) | (0.046) | (0.048) | (0.041) |
| pmcst | | 0.057 *** | | | | |
| | | (0.014) | | | | |
| lands | | | | 0.109 *** | | |
| | | | | (0.017) | | |
| rltion | | | | | | 0.036 * |
| | | | | | | (0.021) |
| Constant | 2.405 * | 7.175 *** | −2.391 ** | 7.901 *** | 8.267 *** | 7.722 *** |
| | (1.228) | (0.572) | (0.990) | (0.660) | (0.701) | (0.601) |
| Control variables | Yes | Yes | Yes | Yes | Yes | Yes |
| Fixed county | Yes | Yes | Yes | Yes | Yes | Yes |
| Observations | 1245 | 1213 | 1610 | 1524 | 1742 | 1322 |
| R-squared | 0.340 | 0.222 | 0.496 | 0.225 | 0.208 | 0.204 |

Note: *** $p < 0.01$, ** $p < 0.05$, * $p < 0.1$.

## 6. Discussion

This study examines the impact of Internet use on land productivity and its mechanisms based on microdata from the 2020 CLES. We find several important results.

First, this study shows that Internet use significantly enhances land productivity, and this finding still holds after a series of robustness tests and endogeneity tests. Combined with the mechanism test, we can see that unlike farm-size IR [43], our conclusion supports the view that China is still in the stage of land productivity increasing with the cultivated land area [49]. The farm-size IR originates from foreign research, but China's national conditions are quite different from those of foreign countries. The characteristics of China's "big country and small farmers" determine that farmers have a small area of cultivated land, and a small area of cultivated land will result in a loss of labor efficiency. The results of the mechanism test show that Internet use can promote farmers to expand the scale of land cultivation, which helps to enhance land productivity. This suggests that improving the capacity of farmers to use digital technologies, such as the Internet, can help to further promote the large-scale management of rural land, thereby enhancing land productivity and achieving sustainable development.

Second, the impact of Internet use on land productivity is heterogeneous across regions, which is similar to the findings of existing studies [53]. Due to the large differences in the structure of economic development in different regions, there are significant differences in farmers' agricultural skills and employment options. This leads to significant differences in the impact of Internet use on land productivity in different regions. According to the empirical results, the Internet use behavior of farmers in the central and northern parts of Jiangsu Province significantly increased land productivity, and the southern part of Jiangsu Province was not significant. Combined with the grain production share of the three major regions in Jiangsu Province in Table 8, it can be seen that the southern region of Jiangsu Province has a low grain production share, and the secondary and tertiary industries in this region are more developed. Therefore, the impact of Internet use on farm households may be mainly in non-agricultural areas.

Finally, mechanism tests show that Internet use has a positive impact on land productivity by expanding farmers' acreage, facilitating mechanized production, and strengthening farmers' social network relationships. Our study supports the idea that the Internet plays a greater role in land transfer in [55]. Land transfer in expands the area of land

cultivated by farmers, which helps to increase land productivity. Internet use helps farmers to mechanize production at a lower cost, e.g., farmers can reduce the cost of mechanized production by purchasing farm machinery services. Mechanized production obviously helps to increase land productivity [6]. Internet use can broaden the social network of farmers [13] and strengthen the connection between farmers and others. This can facilitate financial support for farmers to upgrade their agricultural production equipment, which in turn contributes to land productivity.

## 7. Conclusions and Recommendation

This paper empirically investigates the impact of Internet use on land productivity and its mechanisms using microdata from Jiangsu Province, China. The results show that, first, Internet use increases land productivity by 12.3%. After a series of robustness tests, Internet use still has a positive effect on land productivity. This finding suggests that advancing Internet penetration in developing countries, especially in rural areas, can help improve land productivity and promote sustainable development.

Second, the results of the heterogeneity analysis show that Internet use mainly has a positive effect on land productivity in the central and northern regions of Jiangsu Province. The regression results for the sample from the southern region of Jiangsu Province show that the coefficient for Internet use is $-0.086$ and insignificant. The regression coefficient for the central region is 0.117, but not significant. When county fixed effects are not included, the regression coefficient is significantly positive at the 10% confidence level. The coefficient for Internet use in the northern region is 0.128 and significant at the 5% confidence level. Considering that the Internet usage rate of farmers in Lianyungang is only 0.266, there is still a lot of room to improve land productivity in the future. To further utilize the role of the Internet in improving land productivity, on the one hand, the government should continue to strengthen the construction of communication facilities in areas with low Internet penetration rates, and to bridge the last kilometer of Internet access to the home. On the other hand, the government should enhance farmers' ability to use the Internet, such as strengthening the guidance on Internet use for farmers with lower education and older farmers, and helping farmers to grasp more information on agricultural production.

Third, the results of the mechanistic analysis indicate that the area cultivated by farmers, mechanized production and social network relations play a partly mediating role. Internet use can increase land productivity by expanding farmers' cultivation area, promoting mechanized production and strengthening farmers' social network relations. This provides guidelines for improving land productivity. First, the land transfer system should be improved to provide conditions for farmers to expand their farming area. Second, the level of socialized agricultural machinery services should be further improved to reduce the cost and threshold of mechanized production for farmers. Third, rural financial support should be further strengthened to provide solid financial support for farmers to update their agricultural production techniques.

**Author Contributions:** Methodology, C.W.; Software, X.D.; Formal analysis, J.P. and C.W.; Data curation, X.D.; Writing—original draft, J.P. All authors have read and agreed to the published version of the manuscript.

**Funding:** This research received the funding by the Ministry of Education in China Youth Project of Humanities and Social Sciences (Project No. 22YJC910010) and the Natural Science Foundation of Sichuan Province of China (2023NSFSC1064) for C.W.

**Data Availability Statement:** The original contributions presented in the study are included in the article, further inquiries can be directed to the corresponding author.

**Conflicts of Interest:** The authors declare that they have no conflict of interest to this work.

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
