# Peer review of "The Impact of Internet Use on Land Productivity: Evidence from China Land Economy Survey"

_land, doi:10.3390/land13020262_

Round 1

Reviewer 1 Report

Comments and Suggestions for Authors

1.        The type of this manuscript and the college should be specified on the first page by the authors.

2.        The abstract is too simplistic. The authors should include more numerical results.

3.        In the introduction section, the authors should address two aspects. Firstly, the internal relationship between the internet and land productivity. Secondly, the contribution of this manuscript.

4.        The citations throughout the text are incorrect. Please revise them.

5.        In the literature review section, the number of papers is limited. I recommend the authors to use subheadings to improve the coherence and cohesion of the manuscript. Additionally, I suggest that the authors read and cite the following papers.

Environmental economic geography: Recent advances and innovative development. Unlocking the potential of collaborative innovation to narrow the inter-city urban land green use efficiency gap: Empirical study on 19 urban agglomerations in China. 

6.        The caption of figure 1 is excessively long.

7.        In section 4.3, why did the authors choose these variables? Please provide evidence.

8.        What is the purpose of the Mechanism Test? How does this analysis relate to the previous analyses?

9.        A discussion section is missing.

10.     More numerical results should be included in the conclusion section. Moreover, the policy implications should be proposed based on the results. In the current version, the policy implications are vague.

Author Response

Dear reviewer,

Thank you for your letter and review of our manuscript entitled "The Impact of Internet use on land productivity: Evidence from China Land Economy Survey" (ID: land- 2836960). All these comments are quite valuable and have helped us tremendously in revising and improving the manuscript, as well as providing important guidance for our research. We have carefully studied these suggestions and made careful revisions. The revised parts have been marked in the manuscript in review mode. Thank you again for your comments and suggestions. Our responses to the review comments and revisions to the manuscript are as follows:

Response to Reviewer 1 Comments

Point 1: The type of this manuscript and the college should be specified on the first page by the authors.

Response 1:  We appreciate and agree with the reviewers' comments. In response, we have added information about the type of manuscript and college in the corresponding places as suggested by the experts.

Point 2: The abstract is too simplistic. The authors should include more numerical results.

Response 2: We strongly agree and accept the reviewers' comments. We have optimized the abstract according to the reviewers' suggestions in two aspects. On the one hand, we have added some numerical results. On the other hand, we added and pointed out the research significance of this paper.

Point 3: In the introduction section, the authors should address two aspects. Firstly, the internal relationship between the internet and land productivity. Secondly, the contribution of this manuscript.

Response 3: We strongly agree and accept the reviewers' comments. Based on the experts' suggestions, we have optimized the introduction in four aspects. First, we have added and strengthened the narrative on the intrinsic relationship between Internet use and land productivity. Second, we have added and clarified the contribution of the manuscript. Third, we have condensed and enhanced the content of the introduction. Fourth, we have deleted parts of the text that are not very relevant to the theme of the article.

Point 4: The citations throughout the text are incorrect. Please revise them.

Response 4: We are very grateful to the reviewers for reviewing this paper so meticulously. Based on the reviewers' suggestions, we have carefully corrected the citations throughout the text.

Point 5: In the literature review section, the number of papers is limited. I recommend the authors to use subheadings to improve the coherence and cohesion of the manuscript. Additionally, I suggest that the authors read and cite the following papers.

        Environmental economic geography: Recent advances and innovative development. Unlocking the potential of collaborative innovation to narrow the inter-city urban land green use efficiency gap: Empirical study on 19 urban agglomerations in China.

Response 5: We appreciate and accept the reviewers' comments. Based on the suggestions, we have made three revisions. First, we have carefully read and cited the literature given by the experts and made many useful additions to the literature review of this paper. Second, we have added subheadings to make the manuscript more readable. Third, we have added a discussion of the existing literature and pointed out the importance of this study.

Point 6: The caption of figure 1 is excessively long.

Response 6: We appreciate and accept the comments made by the reviewers. As suggested, we have removed the redundant information in the title of figure 1, which is reflected in the legend.

Point 7: In section 4.3, why did the authors choose these variables? Please provide evidence.

Response 7: We appreciate and accept the reviewers' comments. Based on this suggestion, we have added and described the references on which the selection of the various variables is based in section 4.3.

Point 8: What is the purpose of the Mechanism Test? How does this analysis relate to the previous analyses?

Response 8: We are grateful to the reviewers for their comments and have provided additional clarification on the issue in the Introduction (1.), Theoretical Analysis and Research Hypotheses (3.), and Mechanism Testing (5.5) of the manuscript. The mechanism test is conducted in this paper for three reasons. First, the mechanism test is to further explore the transmission path of Internet use affecting land productivity. Second, the mechanism test is to test the theories and hypotheses in Section 3. Third, the mechanism test is also to propose more feasible, effective and reasonable policy recommendations.

The mechanism test is linked to the previous section in three ways. First, the mechanism test echoes the theoretical analysis in Part 3, and it tests the hypotheses in Part 3. Second, the mechanism test deepens the research on the article's topic, and it provides insight into how Internet use affects land productivity. Third, the mechanism analysis provides useful insights for the final policy recommendations. For example, to improve land productivity in regions with relatively low rates of Internet use, the government should not only directly strengthen the construction of communication infrastructure. At the same time, the government also needs to pay attention to the mechanism variables of Internet use affecting land productivity, so that the positive impact of Internet use on land productivity can be more effectively realized.

Point 9: A discussion section is missing.

Response 9: We are very grateful and agree with the reviewers for such useful comments. In response, we have added a discussion section that takes into account the empirical results of the article. And we have labeled the discussion section in the revised manuscript.

Point 10: More numerical results should be included in the conclusion section. Moreover, the policy implications should be proposed based on the results. In the current version, the policy implications are vague.

Response 10: We appreciate and accept the reviewers' comments. We have made the following three revisions. First, we optimized the original conclusions with the empirical analysis of the article. Second, we added some key numerical results. Third, we propose more targeted policy implications based on the optimized conclusions.

Reviewer 2 Report

Comments and Suggestions for Authors

The use of the Internet by farmers may be important in obtaining information on various factors affecting productivity and income. The proposed research is important. However, the presented results do not bring anything new (at least the authors have not proven it). The statistical analysis used is correct.

Detailed comments:

1. The first hypothesis is too obvious (it is repeated in every subsequent hypothesis)

2. Survey research should be described more detailed.

3. There is no information on general access to the Internet (there is only a variable determining the percentage of farmers using the Internet). I think there should be a relationship here with the farm's income. The higher farmers' income, the greater the percentage of people using the Internet.

4. The literature review is correct and comprehensive, but there is no discussion of the research results. You should add a discussion section and refer to the research of other authors (confirm it or not and show what new results the obtained research brings).

5. It should be indicated what new contributions the obtained research results bring to land productivity.

6. The abstract should define the purpose of the research.

Author Response

Dear reviewer,

Thank you for your letter and review of our manuscript entitled "The Impact of Internet use on land productivity: Evidence from China Land Economy Survey" (ID: land- 2836960). All these comments are quite valuable and have helped us tremendously in revising and improving the manuscript, as well as providing important guidance for our research. We have carefully studied these suggestions and made careful revisions. The revised parts have been marked in the manuscript in review mode. Thank you again for your comments and suggestions. Our responses to the review comments and revisions to the manuscript are as follows:

Response to Reviewer 2 Comments

Point 1: The first hypothesis is too obvious (it is repeated in every subsequent hypothesis).

Response 1: We appreciate and agree with the reviewers' comments. Based on the reviewers' suggestions, we have made the following adjustments to the hypotheses section. On the one hand, we optimized the narrative of the first hypothesis section and highlighted the results of the direct effect of Internet use on land productivity. On the other hand, we corrected part of the narrative of the subsequent hypotheses and highlighted the mechanism by which Internet use indirectly affects land productivity. In fact, Internet use can affect land productivity both directly and indirectly. The first hypothesis emphasizes the direct impact and the last three hypotheses emphasize the indirect impact and the role of the three mediating variables. Indirect impacts refer to the fact that Internet use affects land productivity by expanding the scale of land operations, promoting mechanized production and strengthening the social network relations of farmers.

Point 2: Survey research should be described more detailed.

Response 2: We appreciate and agree with the reviewers' comments. Taking into account the reviewers' suggestions and the information published on the official website of the CLES database, we have added two points to the content. First, we have described the composition of the survey database. Second, we have described the panels included in the farm household questionnaire and the village questionnaire, respectively.

Point 3: There is no information on general access to the Internet (there is only a variable determining the percentage of farmers using the Internet). I think there should be a relationship here with the farm's income. The higher farmers' income, the greater the percentage of people using the Internet.

Response 3: We are very grateful to the reviewers for reviewing this paper so carefully. As far as the core explanatory variables are concerned, on the one hand, we set Internet use as a dummy variable by referring to previous studies [1-4]. On the other hand, in order to provide as much information as possible about Internet use, as well as to enhance the credibility of our findings, we replaced the core explanatory variables in 5.2 in conjunction with the questionnaire. We used the number of smartphones in the household, the presence of an Internet-connected computer at home, and the number of smartphones per capita in households as proxy variables for Internet use, respectively. The regression results support the conclusions of the benchmark regression.

As for farmers' income, we do not think it needs to be included. First, none of the literature studying land productivity controls for farmers' income, but rather for the per capita income of the farmer's village [4-6]( Of course, we also control for this variable.). Second, this paper adds three classes of control variables that can determine the level of farmers' income to an extremely large extent. To minimize the effect of multicollinearity, this manuscript does not include farmers' income as a control variable. This may be the reason for not controlling for farmers' income in the literature.

Point 4: The literature review is correct and comprehensive, but there is no discussion of the research results. You should add a discussion section and refer to the research of other authors (confirm it or not and show what new results the obtained research brings).

Response 4: We are quite grateful and agree with the reviewers' comments. As suggested by the reviewers, we have referred to the studies of other authors and added a discussion section to the literature review. At the same time, we have pointed out the new findings obtained from this study.

Point 5: It should be indicated what new contributions the obtained research results bring to land productivity.

Response 5: We appreciate and agree with the reviewers' comments. As suggested by the reviewers, we have added and pointed out new contributions of this study in several places in the text. Specifically, we have pointed out the contributions of this paper in the Abstract, Discussion, and Conclusions and Recommendations.

Point 6: The abstract should define the purpose of the research.

Response 6: We strongly appreciate and agree with the reviewers' comments. Based on the reviewers' suggestions, we have not only added the purpose of this study in the abstract, but also optimized the significance of the research in this manuscript in the introduction (1.) and the literature review (2.).

References:

  • Carletto, C.; Savastano, S; Zezza, A. Fact or artifact: the impact of measurement errors on the farm size-productivity relationship. Journal of Development Economics, 2013, 103, 254–261.
  • Ma, W.L.; Nie, P.; Zhang, P.; Renwick, A. Impact of Internet use on economic well-being of rural households: Evidence from China. Review of Development Economics. 2020, 24, 503–523.
  • Zheng, H.Y.; Ma, W.L.; Wang, F.; Li, G.C. Does internet use improve technical efficiency of banana production in China? Evidence from a selectivity-corrected analysis. Food Policy. 2021, 102, 102044.
  • Zhou, X.S.; Ma, W.L. Agricultural mechanization and land productivity in China. International Journal of Sustainable Development & World Ecology. 2022, 29, 530–542.
  • Qian, L.; Hong, M.Y. Non-agricultural employment, land transfer and productive efficiency of agriculture: an empirical analysis based on CFPS. Chinese Rural Economy. 2016, 12, 2–16.
  • Muyanga, M.; Jayne, T.S. Revisiting the farm size-productivity relationship based on a relatively wide range of farm sizes: Evidence from Kenya. American Journal of Agricultural Economics. 2019, 101, 1140–1163.

Reviewer 3 Report

Comments and Suggestions for Authors

This study explores the positive impacts and mechanisms of Internet use on land productivity, offering valuable insights for enhancing land productivity.

Interesting work. It is a work that must be continued in terms of sustainability. In the future, farmers' education levels can be added as well as new parameters can be added and updated.

The hypotheses are appropriate in terms of production efficiency in the agricultural field in the use of the Internet.

Why this region was chosen as the study area to realise the hypotheses should be explained in the paper. What is the importance of this region?Emphasizing this will make it easier for readers to understand.

Why did you do Benchmark Regression analysis? Are there no other methods? Is there any clarity at this point in the article? After defining the variables in the article, Benchmark Regression is explained directly.

According to the methods used in the study, 10 tables were produced. However, the results obtained are generalised and presented in the conclusion and recommendations section. The explanations in the conclusion and recommendations section should be supported by giving references to the tables and figures in the article.

The institution from which the data were obtained (CLES) should be acknowledged in an appropriate section of the manuscript.

Author Response

Dear reviewer,

Thank you for your letter and review of our manuscript entitled "The Impact of Internet use on land productivity: Evidence from China Land Economy Survey" (ID: land- 2836960). All these comments are quite valuable and have helped us tremendously in revising and improving the manuscript, as well as providing important guidance for our research. We have carefully studied these suggestions and made careful revisions. The revised parts have been marked in the manuscript in review mode. Thank you again for your comments and suggestions. Our responses to the review comments and revisions to the manuscript are as follows:

Response to Reviewer 3 Comments

Point 1: Why this region was chosen as the study area to realise the hypotheses should be explained in the paper. What is the importance of this region? Emphasizing this will make it easier for readers to understand.

Response 1: We appreciate and agree with the reviewers' comments. Based on the reviewers' suggestions, we have made the following adjustments. On the one hand, we have highlighted the original rationale for choosing Jiangsu Province as the study subject in the first paragraph of 4.1. On the other hand, we have added some of the reasons for choosing the region.

Point 2: Why did you do Benchmark Regression analysis? Are there no other methods? Is there any clarity at this point in the article? After defining the variables in the article, Benchmark Regression is explained directly.

Response 2: We are very grateful to the reviewers for their comments. First, we added the rationale for using the OLS model in 4.4 and gave references for using this method to study land productivity. Second, we have tried to use causal inference methods such as difference-in-differences for empirical testing. However, since the data and region of this study did not fulfill the conditions for using difference-in-differences, we chose the OLS model. Third, theoretically, the result of the benchmark regression obtained using OLS is the best linear unbiased estimator. Thus, benchmark regression is a relatively good method for estimating the effect of Internet use on land productivity.

Point 3: According to the methods used in the study, 10 tables were produced. However, the results obtained are generalised and presented in the conclusion and recommendations section. The explanations in the conclusion and recommendations section should be supported by giving references to the tables and figures in the article.

Response 3: We appreciate and incorporate the reviewers' comments. On the one hand, we have used these results directly in the Discussion (6.), Conclusions and Recommendations (7.) sections. On the other hand, we have combined these results to give more appropriate recommendations.

Point 4: The institution from which the data were obtained (CLES) should be acknowledged in an appropriate section of the manuscript.

Response 4: We strongly appreciate and accept the reviewers' comments. We have checked and corrected the data source organizations in 4.2 and added information about the data.

Round 2

Reviewer 1 Report

Comments and Suggestions for Authors

I agree the publication of this manuscript. Please remind to use track changes to revise your manuscript next time. Thanks.

Reviewer 2 Report

Comments and Suggestions for Authors

I would like to thank the authors for their comprehensive responses to the comments contained in the review. In the current version, I accept the article for publication.